# Adv3D: Generating 3D Adversarial Examples in Driving Scenarios with NeRF

## Abstract

Deep neural networks (DNNs) have been proven extremely susceptible to adversarial examples, which raises special safety-critical concerns for DNN-based autonomous driving stacks (*i.e.*, 3D object detection). Although there are extensive works on image-level attacks, most are restricted to 2D pixel spaces, and such attacks are not always physically realistic in our 3D world. Here we present Adv3D, the first exploration of modeling adversarial examples as Neural Radiance Fields (NeRFs). Advances in NeRF provide photorealistic appearances and 3D accurate generation, yielding a more realistic and realizable adversarial example. We train our adversarial NeRF by minimizing the surrounding objects' confidence predicted by 3D detectors on the training set. Then we evaluate Adv3D on the unseen validation set and show that it can cause a large performance reduction when rendering NeRF in any sampled pose. To generate physically realizable adversarial examples, we propose primitive-aware sampling and semantic-guided regularization that enable 3D patch attacks with camouflage adversarial texture. Experimental results demonstrate that the trained adversarial NeRF generalizes well to different poses, scenes, and 3D detectors. Finally, we provide a defense method to our attacks that involves adversarial training through data augmentation.

## 1 Introduction

The perception system of self-driving cars heavily rely on DNNs to process input data and comprehend the environment. Although DNNs have exhibited great improvements in performance, they have been found vulnerable to adversarial examples [2, 15, 24, 41]. These adversarial examples crafted by adding imperceptible perturbations to input data, can lead DNNs to make wrong predictions. Motivated by the safety-critical nature of self-driving cars, we aim to explore the possibility of generating physically realizable adversarial examples to disrupt 3D detectors in driving scenarios, and further improve the robustness of 3D detectors through adversarial training.

The 2D pixel perturbations (digital attacks) [15, 41] have been proven effective in attacking DNNs in various computer vision tasks [13, 53, 56]. However, these 2D pixel attacks are restricted to digital space and are difficult to realize in our 3D world. To address this challenge, several works have proposed physical attacks. For example, Athalye *et al.* [2] propose the framework of Expectation Over Transformation (EOT) to improve the attack robustness over 3D transformation. Other researchers generate adversarial examples beyond image space through differentiable rendering, as seen in [54, 59]. These methods show great promise for advancing the field of 3D adversarial attacks and defense but are still limited in synthetic environments.

Given the safety-critical demand for self-driving cars, several works have proposed physically realizable attacks and defense methods in driving scenarios. For example, Cao *et al.* [5, 6] propose to learn 3D-aware adversarial attacks capable of generating adversarial mesh to attack 3D detectors.

| Methods | Transferability | Adv. Type | Additional Requirements |
|---------|-----------------|-----------|--------------------------|
| Cao *et al.* [5,6] | Poses | 3D Mesh | Model, Annotation |
| Tu *et al.* [43,44] | Poses, Scenes | 3D Mesh | Model, Annotation |
| Xie *et al.* [57] | Scenes, Categories | 2D Patch | Model, Annotation |
| Adv3D | Poses, Scenes, Categories | 3D NeRF | Model |

Table 1: Comparison with prior works of adversarial attack in autonomous driving.

However, their works only consider learning a 3D adversarial example for a few specific frames. Thus, the learned example is not universal and may not transfer to other scenes. To mitigate this problem, Tu *et al.* [43,44] propose to learn a transferable adversary that is placed on top of a vehicle. Such an adversary can be used in any scene to hide the attacked object from 3D detectors. However, reproducing their attack in our physical world can be challenging since their adversary must have direct contact with the attacked object. We list detailed comparisons of prior works in Tab. 1.

To address the above challenges and generate 3D adversarial examples in driving scenarios, we build Adv3D upon recent advances in NeRF [35] that provide both differentiable rendering and realistic synthesis. In order to generate physically realizable attacks, we model Adv3D in a patch-attack [40] manner and use an optimization-based approach that starts with a realistic NeRF object [26] to learn its 3D adversarial texture. We optimize the adversarial texture to minimize the predicted confidence of all objects in the scenes, while keeping shape unchanged. During the evaluation, we render the input agnostic NeRF in randomly sampled poses, then we paste the rendered patch onto the unseen validation set to evaluate the attack performance. Owing to the transferability to poses and scenes, our adversarial examples can be executed without prior knowledge of the scene and do not need direct contact with the attacked objects, thus making for more feasible attacks compared with [43,44,57,62]. Finally, we provide thorough evaluations of Adv3D on camera-based 3D object detection with the nuScenes [4] dataset. Our contributions are summarized as follows:

- We introduce Adv3D, the first exploration of formulating adversarial examples as NeRF to attack 3D detectors in autonomous driving. Adv3D provides 3D-aware and photorealistic synthesis that was previously unavailable.
- By incorporating the proposed primitive-aware sampling and semantic-guided regularization, Adv3D generates adversarial examples with enhanced physical realism and realizability.
- We conduct extensive real-world experiments and demonstrate the transferability of our adversarial examples across unseen environments and detectors.

## 2 Related Work

### 2.1 Adversarial Attack

DNNs are known to be vulnerable to adversarial attacks, where a small perturbation in the input data can cause drastic changes in the output predictions. Szegedy *et al.* [41] first discovered that adversarial examples, generated by adding visually imperceptible perturbations to the original images, make DNNs predict a wrong category with high confidence. These vulnerabilities were also discovered in object detection and semantic segmentation [30,56]. Moreover, DPatch [30] proposes transferable patch-based attacks by compositing a small patch to the input image. However, perturbing image pixels alone does not guarantee that adversarial examples can be created in the physical world. To address this issue, several works have performed physical attacks [3,8,18,23,46,52,58,61] and exposed real-world threats. For example, Athalye *et al.* [2] generated robust 3D adversarial objects by introducing the Expectation Over Transformation (EOT) method. Cheng *et al.* [11] developed an adversarial patch with physical-oriented transformations to attack a depth estimation network. In our work, we mainly aim to generate 3D adversarial examples for 3D object detection in driving scenarios.

### 2.2 Robustness in Autonomous Driving

With the safety-critical nature, it is necessary to pay special attention to robustness in autonomous driving systems [47]. LiDAR-Adv [6] proposes to learn input-specific adversarial point clouds to fool LiDAR detectors. Tu *et al.* [44] produces generalizable point clouds that can be placed on a

80 vehicle roof to hide it. Furthermore, several work [1, 5, 43] try to attack a multi-sensor fusion system
81 by optimizing 3D mesh through differentiable rendering. We compare our method with prior works
82 in Tab. 1. Our method demonstrates stronger transferability and fewer requirements than prior works.

### 2.3 Image Synthesis using NeRF

84 NeRF [35] enables photorealistic synthesis in a 3D-aware manner. Recent advances [45, 60] in NeRF
85 allow for control over materials, illumination, and 6D pose of objects. Additionally, NeRF's rendering
86 comes directly from real-world reconstruction, providing more physically accurate and photorealistic
87 synthesis than previous mesh-based methods that relied on human handicrafts. Moreover, volumetric
88 rendering [19] enables NeRF to perform accurate and efficient gradient computation compared with
89 dedicated renderers in mesh-based differentiable rendering [9, 21, 29].

90 Recently, there has been tremendous progress in driving scene simulation using NeRF. Block-
91 NeRF [42] achieves city-scale reconstruction by modeling the blocks of cities with several isolated
92 NeRFs to increase capacity. FEGR [51] learns to intrinsically decompose the driving scene for
93 applications such as relighting. Lift3D [26] use NeRF to generate new objects and augment them to
94 driving datasets, demonstrating the capability of NeRF to improve downstream task performance. The
95 driving scene simulation provides a perfect test bed to evaluate the effectiveness of self-driving cars.
96 Our method is related to Lift3D, but aims to understand and improve the robustness of 3D detectors.

## 3 Preliminary

### 3.1 Camera-based 3D Object Detection in Autonomous Driving

99 Camera-based 3D object detection is the fundamental task in autonomous driving. Without loss of
100 generality, we focus on evaluating the robustness of camera-based 3D detectors.

101 The 3D detectors process image data and aim to predict 3D bounding boxes of all surrounding objects.
102 The parameterization of a 3D bounding box can be written as $\mathbf{b} = \{\mathbf{R}, \mathbf{t}, \mathbf{s}, c\}$, where $\mathbf{R} \in SO(3)$ is
103 the rotation of the box, $\mathbf{t} = (x, y, z)$ indicate translation of the box center, $\mathbf{s} = (l, w, h)$ represent the
104 size (length, width, and height) of the box, and $c$ is the confidence of the predicted box.

105 The network structure of camera-based 3D object detectors can be roughly categorized into FoV-
106 based (front of view) and BEV-based (bird's eye view). FoV-based methods [48, 49, 50] can be easily
107 built by adding 3D attribute branches to 2D detectors. BEV-based methods [38, 39] typically convert
108 2D image feature to BEV feature using camera parameters, then directly detect objects on BEV
109 planes. We refer readers to recent surveys [25, 31] for more detail.

### 3.2 Differentiable Rendering using NeRF

111 Our method leverages the differentiable rendering scheme proposed by NeRF [35]. NeRF parame-
112 terizes the volumetric density and color as a function of input coordinates. NeRF uses multi-layer
113 perceptron (MLP) or hybrid neural representations [7, 14, 36] to represent this function. For each
114 pixel on an image, a ray $\mathbf{r}(t) = \mathbf{r}_o + \mathbf{r}_d \cdot t$ is cast from the camera's origin $\mathbf{r}_o$ and passes through the
115 direction of the pixel $\mathbf{r}_d$ at distance $t$. In a ray, we uniformly sample $K$ points from the near plane
116 $t_{near}$ to the far plane $t_{far}$, the $k^{th}$ distance is thus calculated as $t_k = t_{near} + (t_{far} - t_{near}) \cdot k/K$.
117 For any queried point $\mathbf{r}(t_k)$ on the ray, the network takes its position $\mathbf{r}(t_k)$ and predicts the per-point
118 color $\mathbf{c}_k$ and density $\tau_k$ with:

$$(\mathbf{c}_k, \tau_k) = \text{Network}\left(\mathbf{r}(t_k)\right). \tag{1}$$

119 Note that we omit the direction term as suggested by [16]. The final predicted color of each pixel
120 $\mathbf{C}(\mathbf{r})$ is computed by approximating the volume rendering integral using numerical quadrature [34]:

$$\mathbf{C}(\mathbf{r}) = \sum_{k=0}^{K-1} T_k \left(1 - \exp\left(-\tau_k \left(t_{k+1} - t_k\right)\right)\right) \mathbf{c}_k,$$
$$\text{with} \quad T_k = \exp\left(-\sum_{k' < k} \tau_{k'} \left(t_{k'+1} - t_{k'}\right)\right). \tag{2}$$

121 We build our NeRF upon Lift3D [26]. Lift3D is a 3D generation framework that generates photoreal-
122 istic objects by fitting multi-view images synthesized by 2D generative modes [20] using NeRF. The
123 network of Lift3D is a conditional NeRF with additional latent code input, which controls the shape

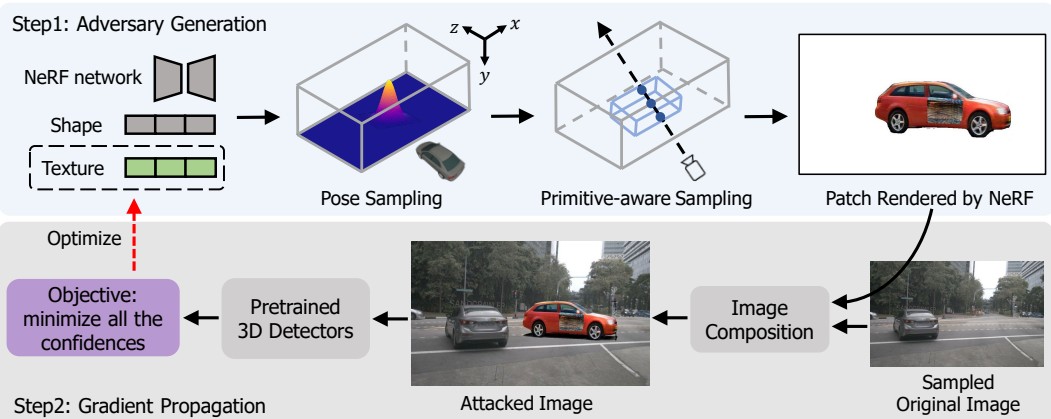

Figure 1: **Adv3D** aims to generate 3D adversarial examples that consistently perform attacks under different poses during rendering. We initialize adversarial examples from Lift3D [26]. During training, we optimize the texture latent codes of NeRF to minimize the detection confidence of all surrounding objects. During inference, we evaluate the performance reduction of pasting the adversarial patch rendered using randomly sampled poses on the validation set.

and texture of the rendered object. The conditional NeRF in Lift3D is a tri-plane parameterized [7] generator. With its realistic generation and 3D controllability, Lift3D has demonstrated that the training data generated by NeRF can help to improve downstream task performance. To further explore and exploit the satisfactory property of NeRF, we present a valuable and important application in this work: we leverage the NeRF-generated data to investigate and improve the robustness of the perception system in self-driving cars.

## 4 Method

We illustrate the pipeline of Adv3D in Fig. 1. We aim to learn a transferable adversarial example in 3D detection that, when rendered in any pose (*i.e.*, location and rotation), can effectively hide surrounding objects from 3D detectors in any scenes by lowering their confidence. In Sec. 4.1, to improve the physical realizability of adversarial examples, we propose (1) Primitive-aware sampling to enable 3D patch attacks. (2) Disentangle NeRF that provides feasible geometry, and (3) Semantic-guided regularization that enables camouflage adversarial texture. To enhance the transferability across poses and scenes, we formulate the learning paradigm of Adv3D within the EOT framework [2] in Sec. 4.3.

### 4.1 3D Adversarial Example Generation

We use a gradient-based method to train our adversarial examples. The training pipeline involves 4 steps: **(i)** randomly sampling the pose of an adversarial example, **(ii)** rendering the example in the sampled pose, **(iii)** pasting the rendered patch into the original image of the training set, and finally, **(iv)** computing the loss and optimizing the latent codes. During inference, we discard the **(iv)** step.

#### 4.1.1 Pose Sampling

To achieve adversarial attack in arbitrary object poses, we apply Expectation of Transformation (EOT) [2] by randomly sampling object poses. The poses of adversarial examples are parameterized as 3D boxes $\mathbf{b}$ that are restricted to a predefined ground plane in front of the camera. We model the ground plane as a uniform distribution $\mathcal{B}$ in a specific range that is detailed in the supplement. During training, we independently sample the rendering poses of adversarial examples, and approximate the expectation by taking the average loss over the whole batch.

#### 4.1.2 Primitive-aware Sampling

We model the primitive of adversarial examples as NeRF tightly bound by 3D boxes, in order to enable non-contact and physically realizable attacks. During volume rendering, we compute the intersection of rays $\mathbf{r}(t)$ with the sampled pose $\mathbf{b} = \{\mathbf{R}, \mathbf{t}, \mathbf{s}\} \in \mathcal{B}$, finding the first hit point and the last hit point of box $(t_{near}, t_{far})$ by the AABB-ray intersection algorithm [33]. We then sample our points inside

the range $(t_{near}, t_{far})$ to reduce large unnecessary samples and avoid contact with the environment.

$$(t_{near}, t_{far}) = Intersect(\mathbf{r}, \mathbf{b}), \tag{3}$$

$$\mathbf{r}'(t_k) = \tilde{\mathbf{r}}(t_{near}) + (\tilde{\mathbf{r}}(t_{far}) - \tilde{\mathbf{r}}(t_{near})) \cdot k/K, \tag{4}$$

$$\tilde{\mathbf{r}}(t) = Transform(\mathbf{r}(t), \mathbf{b}), \tag{5}$$

where $\tilde{\mathbf{r}}(t)$ is the sampled points with additional global to local transformation. Specifically, we use a 3D affine transformation to map original sampled points $\mathbf{r}(t) = \mathbf{r}_o + \mathbf{r}_d \cdot t$ into a canonical space $\tilde{\mathbf{r}} = \{x, y, z\} \in [-1, 1]$. This ensures that all the sampled points regardless of their distance from the origin, are transformed to the range $[-1, 1]$, thus providing a compact input representation for NeRF network. The transformation is given by:

$$Transform(\mathbf{r}, \mathbf{b}) = \mathbf{s}^{-1} \cdot (\mathbf{R}^{-1} \cdot \mathbf{r} - \mathbf{t}), \tag{6}$$

where $\mathbf{b} = \{\mathbf{R}, \mathbf{t}, \mathbf{s}\}$, $\mathbf{R} \in SO(3)$ is rotation matrix of the box, $\mathbf{t}, \mathbf{s} \in \mathbb{R}^3$ indicate translation and scale vector that move and scale the unit cube to desired location and size. The parameters of $\mathbf{b}$ are sampled from a pre-defined distribution $\mathcal{B}$ detailed in the supplement.

Then, the points lied in $[-1, 1]$ are projected to exactly cover the tri-plane features $\mathbf{z}$ for interpolation. Finally, a small MLP takes the interpolated features as input and predicts RGB and density:

$$(\mathbf{c}_k, \tau_k) = \text{MLP}(Interpolate(\mathbf{z}, \mathbf{r}'(t_k))). \tag{7}$$

The primitive-aware sampling enables patch attacks [40] in a 3D-aware manner by lifting the 2D patch to a 3D box, enhancing the physical realizability by ensuring that the adversarial example only has a small modification to the original 3D environment.

### 4.1.3 Disentangled NeRF Parameterization

The original parameterization of NeRF combines the shape and texture into a single MLP, resulting in an entangled shape and texture generation. Since shape variation is challenging to reproduce in the real world, we disentangle shape and texture generation and only set the texture as adversarial examples.

We obtain texture latents $\mathbf{z_{tex.}}$ and shape latents $\mathbf{z_{shape}}$ from the Lift3D. During volume rendering, we disentangle shape and texture generation by separately predicting RGB and density:

$$\mathbf{c}_k = \text{Network}(\mathbf{z_{tex.}}, \mathbf{r}'(t_k)), \quad \tau_k = \text{Network}(\mathbf{z_{shape}}, \mathbf{r}'(t_k)), \tag{8}$$

where $\mathbf{z_{shape}}$ is fixed and $\mathbf{z_{texture}}$ is being optimized. Our disentangled parametrization can also be seen as a geometry regularization in [43, 44] but keeps geometry unchanged as a usual vehicle, leading to a more realizable adversarial example.

### 4.1.4 Semantic-guided Regularization

Setting the full part of the vehicle as adversarial textures is straightforward, but not always feasible in the real world. To improve the physical realizability, we propose to optimize individual semantic parts, such as doors and windows of a vehicle. Specifically, as shown in Fig. 2 (d, e)), we only set a specific part of the vehicle as adversarial texture while maintaining others unchanged. This semantic-guided regularization leads to a camouflage adversarial texture that is less likely spotted in the real world.

To achieve this, we add a semantic branch to Lift3d [26] to predict semantic part labels of the sampled points. We re-train Lift3d by fitting multi-view images and semantic labels generated by editGAN [28]. Using semantic-guided regularization, we maintain the original texture and adversarial part texture at the same time but only optimize the adversarial part texture while leaving the original texture unchanged. This approach allows us to preserve a large majority of parts as usual, but to alter only the specific parts that are adversarial (see Fig. 2 (b, c)). Potential attackers can easily print the adversarial sticker and stick it on the semantic part of vehicles to hide surrounding objects.

In our implementation, we query the NeRF network twice, one for the adversarial texture and the other for the original texture. Then, we replace the part of original texture with the adversarial texture indexed by semantic labels in the point space.

## 4.2 Gradient Propagation

After rendering the adversarial examples, we paste the adversarial patch into the original image through image composition. The attacked image can be expressed as $I_1 \times M + I_2 \times (1 - M)$ where $I_1$ and $I_2$ are the patch and original image, $M$ is foreground mask predicted by NeRF. Next, the attacked images are fed to pretrained and fixed 3D detectors to compute the objective and back-propagate the gradients. Since both the rendering and detection pipelines are differentiable, Adv3D allows gradients from the objective to flow into the texture latent codes during optimization.

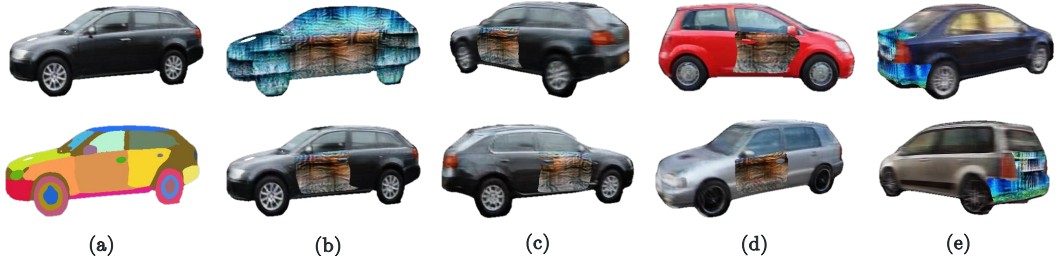

(a)     (b)     (c)     (d)     (e)

Figure 2: Rendered results of our adversarial examples. **(a)** Image and semantic label of an instance predicted by NeRF. **(b)** Top: our example without semantic-guided regularization. Bottom: our example with semantic-guided regularization. **(c)** Multi-view consistent synthesis of our examples. **(d,e)** The texture transfer results of side and back part adversary to other vehicles.

## 4.3 Learning Paradigm

We formulate our learning paradigm as EOT [2] that finds adversarial texture latent codes by minimizing the expectation of a binary cross-entropy loss over sampled poses and original images:

$$\mathbf{z_{tex.}} = \arg\min_{\mathbf{z_{tex.}}} \mathbb{E}_{\mathbf{b} \sim \mathcal{B}} \mathbb{E}_{\mathbf{x} \sim \mathcal{X}} [-\log(1 - P(I(\mathbf{x}, \mathbf{b}, \mathbf{z_{tex.}})))], \tag{9}$$

where $\mathbf{b}$ is the rendering pose sampled from the predefined distribution of ground plane $\mathcal{B}$, $\mathbf{x}$ is the original image sampled from the training set $\mathcal{X}$, $I(\mathbf{x}, \mathbf{b}, \mathbf{z_{tex.}})$ is the attacked image that composited by the original image $\mathbf{x}$ and the adversarial patch rendered using pose $\mathbf{b}$ and texture latent code $\mathbf{z_{tex.}}$, and $P(I(\cdot))$ represents the confidence of all proposals predicted by detectors. We approximate the expectation by averaging the objective of the independently sampled batch. The objective is a binary cross-entropy loss that minimizes the confidence of all predicted bounding boxes, including adversarial objects and normal objects.

Built within the framework of EOT, Adv3D helps to improve the transferability and robustness of adversarial examples over the sampling parameters (poses and scenes here). This means that the attack can be performed without prior knowledge of the scene and are able to disrupt models across different poses and times in a non-contact manner.

## 4.4 Adversarial Defense by Data Augmentation

Toward defenses against our adversarial attack, we also study adversarial training to improve the robustness of 3D detectors. Adversarial training is typically performed by adding image perturbations using a few PGD steps [32, 55] during the training of networks. However, our adversarial example is too expensive to generate for the bi-level loop of the min-max optimization objective. Thus, instead of generating adversarial examples from scratch at every iteration, we directly leverage the transferable adversarial examples to augment the training set. We use the trained adversarial example to locally store a large number of rendered images to avoid repeated computation. During adversarial training, we randomly paste the rendered adversarial patch into the training images with a probability of 30%, while remaining others unchanged. We provide experimental results in Sec. 5.4.

| Models | Backbone | Type | Clean NDS | Adv NDS | Clean mAP | Adv mAP |
|---|---|---|---|---|---|---|
| FCOS3D [48] | ResNet101 | FoV | 0.3770 | 0.2674 | 0.2980 | 0.1272 |
| PGD-Det [49] | ResNet101 | FoV | 0.3934 | 0.2694 | 0.3174 | 0.1321 |
| DETR3D [50] | ResNet101 | FoV | 0.4220 | 0.2755 | 0.3470 | 0.1336 |
| BEVDet [17] | ResNet50 | BEV | 0.3822 | 0.2247 | 0.3076 | 0.1325 |
| BEVFormer-Tiny [27] | ResNet50 | BEV | 0.3540 | 0.2264 | 0.2524 | 0.1217 |
| BEVFormer-Base [27] | ResNet101 | BEV | 0.5176 | 0.3800 | 0.4167 | 0.2376 |

Table 2: Comparison of different detectors under our attack. Clean NDS and mAP denote evaluation using original validation data. Adv NDS and mAP denote evaluation using attacked data.

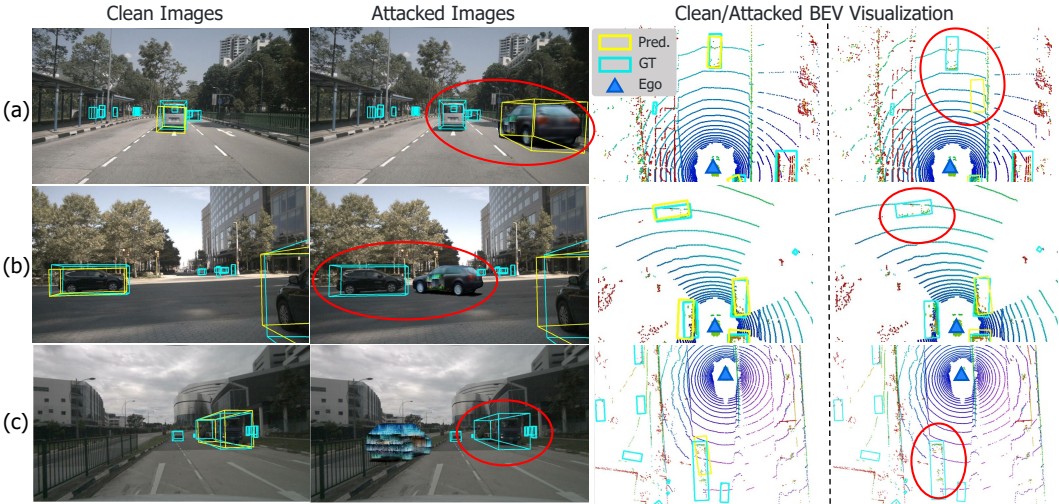

Figure 3: Visualization of BEVDet [17] prediction on nuScenes validation set under our attacks. The visualization threshold is set at 0.6. The adversarial NeRF can hide surrounding objects by minimizing their predicted confidence in a non-contact manner (making the yellow boxes disappear).

## 5 Experiments

In this section, we first describe the training detail of our adversarial attacks. Then we present the experiments of semantic-guided regularization in Sec. 5.1, the analysis of 3D attack in Sec. 5.2, the transferability across detectors in Sec. 5.3, and our adversarial defense method in Sec. 5.4.

**Dataset**     We conduct our experiments on the widely used nuScenes dataset [4]. This dataset is collected using 6 surrounded-view cameras that cover the full 360° field of view around the ego-vehicle. The dataset contains 700 scenes for training and 150 scenes for validation. In our work, we train our adversarial examples on the training set and evaluate performance drop on the validation set.

**Training**     We implement our methods using PyTorch [37] and MMDetection3D [12]. All detectors are resumed from checkpoints available on their open-source repositories to match the original performance exactly. We only select one instance from Lift3D [26] as the initialization of examples. We conduct our experiments using 8 NVIDIA A100 80G GPUs. We use the Adam optimizer [22] with a learning rate of 1e-3 for texture latents. In practice, we optimize texture latents on the training set for five epochs with the same batch size as used during training detectors. We do not use any regularization except for semantic-guided regularization. In all experiments without specified, we render two adversarial examples per image. We ablate the number of rendered adversaries in the supplement.

**Target Detectors and Metrics**     We evaluate the robustness of six representative detectors. Three are FoV-based, and three are BEV-based. The FoV-based methods are FCOS3D [48], PGD-Det [49] and DETR3D [50]. The BEV-based methods are BEVDet [17], BEVFormer-Tiny [27] and BEVFormer-Base. Following prior work [57], we evaluate the performance drop on the validation set after the attack. Specifically, we use the Mean Average Precision (mAP) and nuScenes Detection Score (NDS) [4] to evaluate the performance of 3D detectors.

**Quantitative Results**     We provide the experimental results of adversarial attacks in Tab. 2. The attacks are conducted in a full-part manner without semantic-guided regularization to investigate the upper limit of attack performance. We found that, in spite of FoV-based or BEV-based, they display similar robustness. Meanwhile, we see a huge improvement of robustness by utilizing a stronger backbone (ResNet101 versus ResNet50) when comparing BEVFormer-Base with BEVFormer-Tiny. We hope these results will inspire researchers to develop 3D detectors with enhanced robustness.

**Visualization Results**     We visualize our attack results with semantic-guided regularization in Fig. 3 (a,b), and without regularization in Fig. 3 (c). The disappearance of detected objects is caused by their lower confidence scores. For example, the confidence predicted by detectors in Fig. 3 (a) have declined

| Semantic Part | NDS | mAP |
|---|---|---|
| Clean | 0.382 | 0.307 |
| No Part | 0.302 | 0.234 |
| Full Parts | 0.224 | 0.132 |
| Part of Front | 0.267 | 0.148 |
| Part of Side | 0.265 | 0.149 |
| Part of Rear | 0.268 | 0.151 |

Table 3: Attack results of different semantic parts.

| Data | Adv train | NDS | mAP |
|---|---|---|---|
| Clean val | ✗ | 0.304 | 0.248 |
| Clean val | ✓ | **0.311** | **0.255** |
| Adv val † | ✗ | 0.224 | 0.132 |
| Adv val † | ✓ | **0.264** | **0.181** |
| Adv val § | ✓ | 0.228 | 0.130 |

Table 4: Results of adversarial training.

from $0.6$ to $0.4$, and are therefore filtered out by the visualization threshold of $0.6$. In Fig. 3 (a), we find that our adversarial NeRF is realistic enough to be detected by a 3D detector if it doesn't display much of the adversarial texture. However, once the vehicle shows a larger area of the adversarial texture as seen in Fig. 3 (b), it will hide all objects including itself due to our untargeted objective.

## 5.1 Semantic Parts Analysis

In Tab. 3, we provide experiments on the impact of different semantic parts on attack performance. Specifically, we focused on three salient parts of the vehicle: the front, side, and rear. Our results show that compared with adversarial attacks using full parts, the semantic-guided regularization leads to a slightly lower performance drop, but remains a realistic appearance and less likely spotted adversarial texture as illustrated in Fig. 2 (b).

Since we do not have access to annotation during training, we additionally conduct "No Part" experiment that no part of the texture is adversarial, to evaluate the impact of the collision and occlusion. We acknowledge that part of performance degradation can be attributed to the occlusion to original objects and the false positive prediction of adversarial objects (see Fig. 3 (a)), since we do not update the ground truth of adversarial objects to the validation set.

## 5.2 Effectiveness of 3D-aware attack

To validate the effectiveness of our 3D attacks, we ablate the impact of different poses on the attack performance. In Fig. 4 (a), we divide the BEV plane into $10 \times 10$ bins ranging from $x \in [-5m, 5m]$ and $z \in [10m, 15m]$. We then evaluate the relative mAP drop (percentage) of BEVDet [17] by sampling one adversarial example inside the bin per image, while keeping rotation randomly sampled. Similarly, we conduct experiments of $30$ uniform rotation bins ranging from $[0, 2\pi]$ in Fig. 4 (b). The experimental results demonstrate that all aspects of location and rotation achieve a valid attack (performance drop $> 30\%$), thereby proving the transferability of poses in our 3D-aware attack.

A finding that contrasts with prior work [44] is the impact of near and far locations in $z$ axis. Our adversarial example is more effective in the near region compared with the far region, while Tu *et al.* [44] display a roughly uniform distribution in all regions. We hypothesize that the attack performance is proportional to the area of the rendered patch, which is highly related to the location

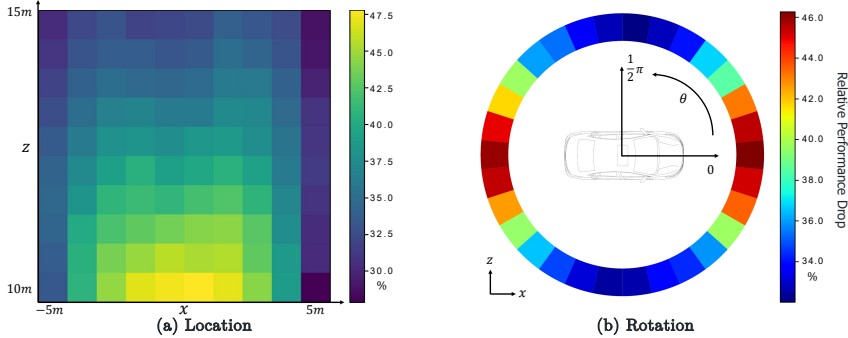

Figure 4: To examine the 3D-aware property of our adversarial examples, we ablate the relative performance drop by sampling adversarial examples within different bins of location and rotation.

| Source Target | Clean | FCOS3D | PGD-Det | DETR3D | BEVDet | BEVFormer |
|---|---|---|---|---|---|---|
| FCOS3D [48] | 0.298 | **0.124** | 0.141 | 0.144 | 0.176 | 0.158 |
| PGD-Det [49] | 0.317 | 0.172 | **0.131** | 0.150 | 0.186 | 0.172 |
| DETR3D [50] | 0.347 | 0.188 | 0.170 | **0.133** | 0.212 | 0.198 |
| BEVDet [17] | 0.307 | 0.148 | 0.145 | 0.140 | **0.132** | 0.140 |
| BEVFormer [27] | 0.252 | 0.175 | 0.155 | 0.136 | 0.177 | **0.124** |

Table 5: Transferability of our attack to unseen detectors. We evaluate the robustness of **target** detectors using an adversarial example trained on **source** detectors. Reported in mAP.

of objects. Similar findings are also displayed in rotation. The vehicle that poses vertically to the ego vehicle results in a larger rendered area, thus better attack performance.

### 5.3 Transferability Across Different Detectors

In Tab. 5, we evaluate the transferability of adversarial examples across different detectors. To this end, we train a single adversarial example of each detector separately, then use the example to evaluate the performance drop of other detectors. We show that there is a high degree of transferability between different models. Among them, we observe that DETR3D [50] appears to be more resilient to adversarial attacks than other detectors. We hypothesize this can be attributed to the sparsity of the query-based method. During the projection of 3D query to the 2D image plane, only a single point of the feature is indexed by interpolation, thus fewer areas of adversarial features will be sampled. This finding may have insightful implications for the development of more robust 3D detectors in the future.

### 5.4 Adversarial Defense by Data Augmentation

We present the results of adversarial training in Tab. 4. The symbol † indicates attacks using the same adversarial example used in adversarial training, while § indicates a different example. We observe that incorporating adversarial training improves not only the robustness against the seen adversarial examples, but also the clean performance. However, we also note that our adversarial training is not capable of transferring to unseen adversarial examples trained in the same way, mainly due to the fixed adversarial example during adversarial training. Furthermore, we hope that future work can conduct in-depth investigations and consider handling the bi-level loop of adversarial training in order to better defend against adversarial attacks.

## 6 Limitation and Future Work

**Learning to Sample and Attack** As we do not have access to the dataset annotations, we can not model the explicit relationship between adversarial and normal objects to avoid collision, and the collision itself can cause a performance drop ("No Parts" in Tab. 3). Future work can apply geometry-aware composition [10] to mitigate this problem. Additionally, future research can explore learning to predict optimal poses of adversarial objects to maximize the effectiveness of attacks.

**Potential Harmful Consequences** The trained adversarial examples have the potential to induce serious traffic accidents in driving scenarios. However, our work is not intended to cause disruptions in autonomous driving systems. Instead, our goal is to use the examples to gain a deeper understanding of the systems and improve their robustness. We hope our work will draw more attention of the community to further verify and enhance the robustness of autonomous driving systems.

## 7 Conclusion

In this paper, we propose **Adv3D**, the first attempt to model adversarial examples as NeRF. Adv3D enhances the physical realizability of attacks through our proposed primitive-aware sampling and semantic-guided regularization. Compared with prior works of adversarial examples in autonomous driving, our examples are more threatening in practice as we carry non-contact attacks, have feasible 3D shapes as usual vehicles, and display camouflage adversarial texture. Extensive experimental results also demonstrate that Adv3d transfers well to different poses, scenes, and detectors. We hope our work provides valuable insights for creating more realistic evaluations to investigate and improve the robustness of autonomous driving systems.

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
