# Supplementary Material for
# Adv3D: Generating 3D Adversarial Examples in Driving Scenarios with NeRF

In this supplementary material, we first detail the sampling pose distribution $\mathcal{B}$ in Sec. A, then we provide analysis and ablation studies on the number of adversarial examples in Sec. B, evaluate the transferability of adversarial texture to the unseen vehicle in Sec. C, and provide repeatability test in Sec. D. Finally, we present additional qualitative results in Sec. E. In our experiments in the supplementary material, we use BEVDet [1] as the attacked 3D detector.

## A   Sampling Pose Distribution

In this section, we introduce the distribution of pose $\mathcal{B}$ used during pose sampling. Specifically, we parameterize the pose $\mathcal{B}$ as 3D bounding boxes that are described by $(x, y, z, l, w, h, \theta)$, where $x, y, z$ is the position of the 3D bounding box's center, $l, w, h$ represent length, width, height of the box, $\theta$ is the rotation along $y$ axis. As shown in Tab. 3, we model the components of $\mathcal{B}$ as independently sampled parameters. We mainly follow the parametrization in Lift3D [2].

To avoid inappropriate sampling in driving scenes (such as a vehicle flying in the sky), we model our objects to roughly be placed on a ground plane in front of the camera. To approximate the ground plane, we set $(x, z)$ to be uniformly sampled from $[-5m, 5m]$ and $[10m, 15m]$. As for $y$, we first obtain camera height from the camera extrinsic, then we model $y$ as a Gaussian distribution that is centered at the camera height. Similarly, $l_{mean}$, $w_{mean}$, and $h_{mean}$ is the mean value of length, width, and height of the 3D box obtained from the statistic of datasets, where $l_{mean} = 4.61m, w_{mean} = 1.95m, h_{mean} = 1.72m$. Then we model the size of 3D boxes as a Gaussian distribution that is centered at the mean value. The rotation of the object is considered as a bimodal distribution that is centered at forward facing($\pi/2$) and backward facing($-\pi/2$).

## B   Ablation of The Number of Adversarial Examples

Since the number of populated adversarial examples is related to the area of adversarial texture, we ablate the number of rendered adversarial examples per image in Tab. 5. We observe that the more adversarial objects, the better the attack performance. The number 0 denotes a clean evaluation without attacks. In all experiments in our main paper except Sec. 5.2, we render two adversarial examples for evaluation to avoid overcrowded scenes. We note that for number $> 1$, the rendered adversarial patches are from the same adversarial example, since we only maintain a single adversarial object during training.

| Experiments | NDS | mAP |
|---|---|---|
| Clean | 0.3822 | 0.3076 |
| SUV | 0.2209 | 0.1310 |
| SUV to Sedan | 0.2344 | 0.1449 |
| SUV to Hatchback | 0.2319 | 0.1413 |
| SUV to Jeep | 0.2348 | 0.1452 |

Table 2: Results of our adversarial texture transfer to unseen shape.

| Pose | Distribution | Parameters |
|---|---|---|
| $x$ | Uniform | $[-5m, 5m]$ |
| $y$ | Gaussian | $\mu = height, \sigma = 0.2$ |
| $z$ | Uniform | $[10m, 15m]$ |
| $l$ | Gaussian | $\mu = l_{mean}, \sigma = 0.5$ |
| $w$ | Gaussian | $\mu = w_{mean}, \sigma = 0.5$ |
| $h$ | Gaussian | $\mu = h_{mean}, \sigma = 0.5$ |
| $\theta$ | Gaussian | $\mu = \pm\pi/2, \sigma = \pi/2$ |

Table 3: Detailed distribution of pose **b** in pose sampling.

| Experiments | NDS | mAP |
|---|---|---|
| Clean | 0.3822 | 0.3076 |
| Trial 1 | 0.2261 | 0.1322 |
| Trial 2 | 0.2263 | 0.1326 |
| Trial 3 | 0.2209 | 0.1310 |
| Trial 4 | 0.2237 | 0.1302 |
| Trial 5 | 0.2228 | 0.1304 |
| Mean | 0.2239 | 0.1313 |
| Std | 0.0023 | 0.0011 |

| Number | NDS | mAP |
|---|---|---|
| 0 | 0.3822 | 0.3076 |
| 1 | 0.2648 | 0.1895 |
| 2 | 0.2247 | 0.1325 |
| 3 | 0.1749 | 0.0698 |

Table 4: Results of repeatability test.

Table 5: Ablation of the number of adversarial examples.

## C  Adversarial Texture Transfer to Unseen Vehicle

To further validate the transferability of our method, we analyze the attack performance of a trained adversarial texture transfer to an unseen shape. As shown in Tab. 2, we train our adversarial examples using only an SUV-type vehicle and then transfer the adversarial texture to other unseen vehicles by our proposed disentangled texture and shape generation. The target types of vehicles include Sedan, Hatchback, and Jeep. The transfer results of the unseen types of vehicles show comparable attack performance compared with the seen one, which demonstrates the transferability of our adversarial texture to unseen shapes. The rendered images of adversarial examples can be found in Fig. 2-4.

## D  Repeatability Test

In our pose sampling, we randomly and independently sample the parameters of the pose for each frame from pose distribution $\mathcal{B}$. In this section, we investigate the impact of the randomness of pose sampling. We conduct five independent evaluations using the same trained adversarial example and show results in Tab. 4. We found that regardless of the random pose sampling, all the trials display a similar attack performance. The results indicate that our adversarial attack is robust against the randomness of pose sampling.

## E  Additional Qualitative Results

In this section, we present additional qualitative results of our trained adversarial example. In Fig. 1, we show rendered images of an SUV-type adversarial example, and different parts of semantic-guided regularization. Then we display rendered images of adversarial texture transfer to the unseen vehicles in Fig. 2-4.

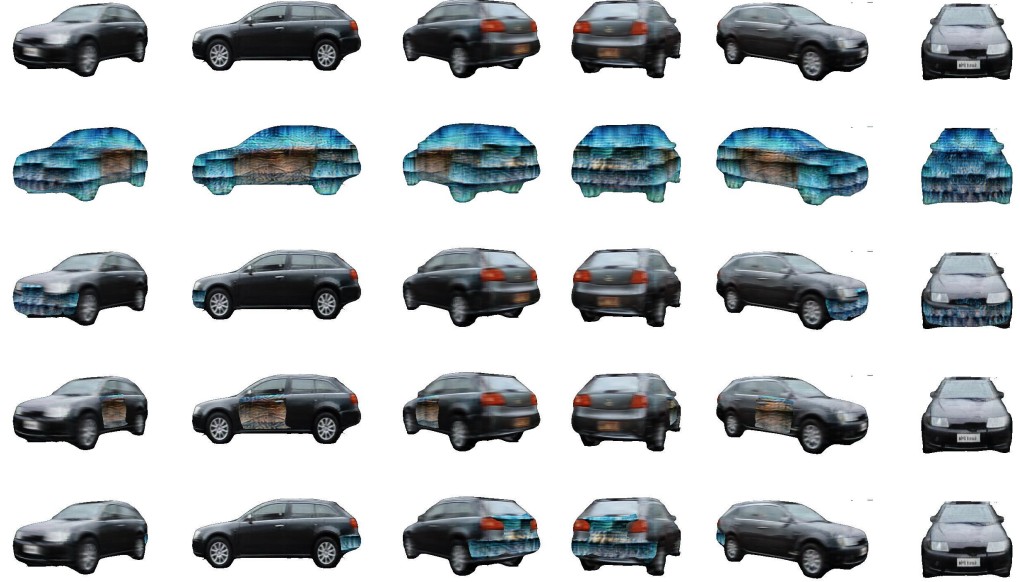

Figure 1: **First row:** the initial SUV-type vehicle. **Second row:** the trained full-part adversarial example using the SUV-type vehicle. **Third row to the fifth row:** The results of full part adversarial texture transfer to the front part, side part, and rear part.

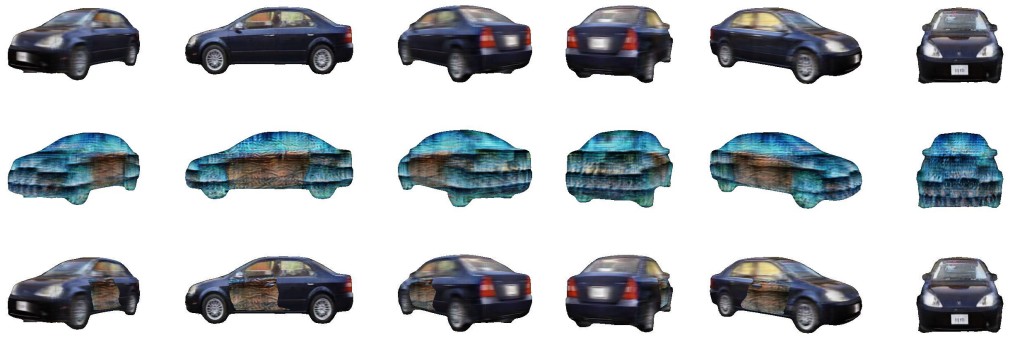

Figure 2: **First row:** Rendered images of a Sedan-type vehicle. **Second and third rows:** We transfer the trained adversarial texture of an SUV-type vehicle (Fig. 1) to the Sedan-type vehicle.

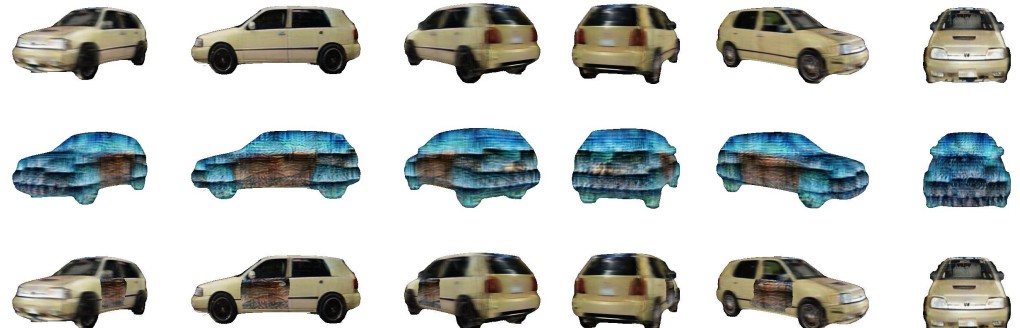

Figure 3: **First row:** Rendered images of a Hatchback-type vehicle. **Second and third rows:** We transfer the trained adversarial texture of an SUV-type vehicle (Fig. 1) to the Hatchback-type vehicle.

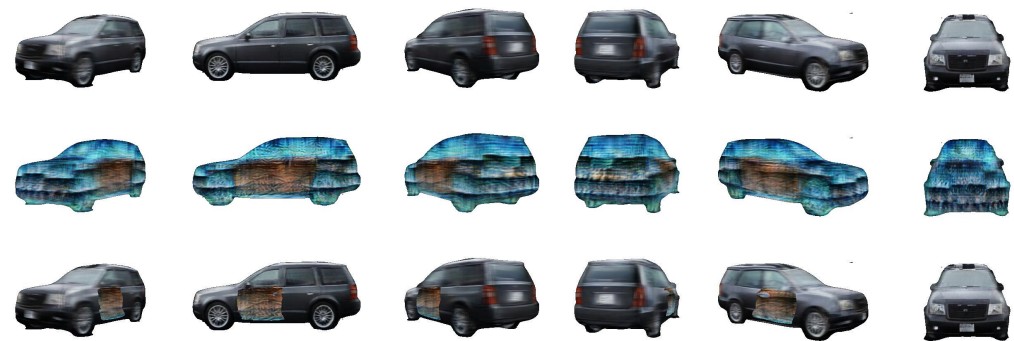

Figure 4: **First row:** Rendered images of a Jeep-type vehicle. **Second and third rows:** We transfer the trained adversarial texture of an SUV-type vehicle (Fig. 1) to the Jeep-type vehicle.