# OpenReview forum: "Adv3D: Generating 3D Adversarial Examples in Driving Scenarios with NeRF"
_NeurIPS.cc/2023/Conference — Submitted to NeurIPS 2023_

### Official Review · Reviewer_s7Sz · 2023-07-06

**Soundness:** 2 fair
**Presentation:** 4 excellent
**Contribution:** 3 good
**Rating:** 7
**Confidence:** 5

**Summary:**

This work proposes a new attack on monocular 3D object detectors utilising NERF representation to create multi-view consistent attacks utilising Lift3D [26]. The authors show successful attacks on nuScenes dataset and the effect of their design choices on the attack success. Furthermore, a mitigating strategy is shown how to make monocular 3D object detection robust to these types of attacks.

**Strengths:**

This manuscript includes a comprehensive set of ablation studies and detailed analyses, which effectively highlight the influence of different components in the proposed pipeline. Figure 4 is especially illustrative and insightful. Notably, the successful tackling of numerous molecular 3D detectors, each with varying detection strategies, is commendable.

The authors present an effective strategy to counteract the proposed adversarial attacks on molecular detection systems.

The overall presentation of the paper is lucid and the figures contribute significantly to accurately conveying the intended ideas.

**Weaknesses:**

1- A comparison against baselines is missing, most notably the mesh-based baseline [43]. This comparison is very important since the proposed setup in Lift3D [26] is very similar when constraining the Nerf to shape and texture latents , resembling the mesh generation and texturing scheme in [43].

2- A crucial evaluation protocol has been overlooked. In the context of adversarial attacks, the imperceptibility of the attack is a significant factor; how discernible is the attack in contrast to the clean sample? This critical information is absent in this work. It leaves us questioning the perceptibility of image corruption in terms of pixel alteration. If the corruption is so pronounced that even a human observer fails to detect the cars, can it still be considered a successful attack? Previous studies typically provide this information [43]. Moreover, the inclusion of tests on KITTI should be considered vital to the evaluation process.

**Questions:**

Despite some concerns, the work presents an innovative methodology for crafting adversarial attacks on the crucial application of monocular 3D object detection in autonomous vehicles. The manuscript is well-structured and successfully demonstrates attacks, supported by a wealth of insightful analysis. However, the absence of comparisons with similar baselines and the lack of detailed evaluation metrics assessing imperceptibility prevent me from fully endorsing the paper at this stage. I would appreciate further clarification from the authors on these points before finalising my decision.

Minor Comments:

The authors could consider including additional related references [a,b] that were overlooked. An animated presentation of a KITI video demonstrating successful attacks would be an informative addition.

[a] AdvPC: Transferable Adversarial Perturbations on 3D Point Clouds, ECCV 2020

[b] SADA: Semantic Adversarial Diagnostic Attacks for Autonomous Applications, AAAI 2020

---------------------------------------------
post rebuttal thoughts :

The authors have addressed my concerns regarding the baselines and the imperceptibility of the attack in the rebuttal. I dont think real-life applications are necessary here ( as other reviewers think) and hence I increase my score to accept

---

> ### Author Rebuttal · Authors · 2023-08-06
>
> We would like to thank the reviewer for the detailed feedback, and for appreciating that our work is innovative, well-structured and provide insightful analysis.
>
> Below, we reply to individual questions and comments raised by the reviewer:
>
> **(1) Mesh Comparison.**
>
> This is a good point. We add an experiment using a ShapeNet car model as a mesh baseline. We use PyTorch3D’s differentiable renderer and optimized the vertex color as an adversarial example to attack 3D detectors (BEVDet). Just like the setting of the NeRF counterpart, we randomly render the mesh model and paste the patch onto the original images. The attack performance of mAP and NDS is slightly lower than the NeRF counterpart. This may be attributed to the latent space of NeRF weights has a higher dimensional representation than vertex color and providing much more solutions for attacking, which results in a better attack performance.
> | Method      | NDS | mAP |
> | ----------- | ----------- | ----------- |
> | Clean      | 0.3822 | 0.3076 |
> | Mesh attack   | 0.3018 | 0.2183 |
> | NeRF attack   | 0.2648 | 0.1895 |
>
>
> **(2) Perceptibility.**
>
> Thank you for bringing this valuable feedback. Compared with [43,44], our adversarial examples are less likely to be spotted by humans, as we carry non-contact attacks, have feasible 3D shapes as usual vehicles, and display camouflage adversarial texture. In addition, in Fig. 3 (a) of the main paper, we demonstrate that our adversarial example is realistic enough to be detected by 3D detectors, as well as by human eyes. We also add statistics investigating the influence of distance and pixel proportion. In the setting of 3D vision, the farther the distance, the smaller the pixel proportion (size of patch / (1600*900)). From the below table, we can observe that as the distance increases, the pixel proportion decreases and the attack performance relatively decreases as well. This indicates that the higher the perceptibility (larger pixel proportion), the better the attack performance.
>
> | Distance (m) | 10 | 11 | 12 | 13 | 14 |
> | ----------- | ----------- | ----------- | ----------- | ----------- | ----------- |
> | Pixel proportion (%)| 4.67 | 3.67 | 2.96 | 2.42 | 2.04 |
> | Performance drop (%)| 41.47 | 39.56 | 37.11 | 34.74 | 32.52 |
>
> **(3) Additional References.**
>
> Thank you for your advice. We will revise the Related Work section to include a discussion of these relevant references.

---

> > ### Comment · Reviewer_s7Sz · 2023-08-14
> >
> > I would like to thank the authors for the great rebuttal and effort for real-world example. I increased my score to 7.

---

> > > ### Author Response · Authors · 2023-08-14
> > > **Thanks for your positive feedback**
> > >
> > > We are glad that we have addressed your concerns. We thank you again for the valuable comments.

---

### Official Review · Reviewer_RYj2 · 2023-07-07

**Soundness:** 3 good
**Presentation:** 2 fair
**Contribution:** 3 good
**Rating:** 5
**Confidence:** 4

**Summary:**

This work develops adversarial attacks against 3D object detectors by utilizing instance-level NeRFs.
They start with a representation of a vehicle, parameterized by a NeRF that predicts both geometry and texture and render the vehicle into a image, which they compose into the original image by copy-pasting. They use the composited image to adversarially attack 3D object detectors, which provides a gradient signal used to optimize the NeRF (texture only).
Experiments show their adversarial examples are effective against a variety of different 3D object detectors, and they show that training on these samples improves robustness (and even overall performance).

**Strengths:**

* Novel application of NeRFs, utilizing the fully differentiability to optimize for adversarial texture.
* Work is well written and overall clear to follow.
* Analysis provides good insights (referring to Sec 5.3 analysis of 3D detector architecture robustness and Sec 5.4 adversarial training actually boosts performance).
* Multiple architectures used in experiments.

**Weaknesses:**

* Section 4.4 could use more elaboration - this is a key section for the overall work and in the current revision is quite vague.
* Some of the attacks (Fig. 3) do not look photorealistic - how can the reader be convinced these adverarial samples would actually work in the real world?

**Questions:**

* Section 4.4: How many rendered images are used? Are there any heuristics used to make sure the pasted patches are physically realizable?
* What are the computational costs of generating a single attack?

**Limitations:**

Authors provide discussion of limitations (real world safety).

---

> ### Author Rebuttal · Authors · 2023-08-06
>
> We would like to thank the reviewer for the detailed feedback and for appreciating the novelty of our idea and clear writing. In the following, we reply to individual questions and comments raised by the reviewer:
>
> **(1) Elaborate Section 4.4.**
>
> In our adversarial training, we first infer the trained adversarial example to locally store 10,000 rendered images to avoid repeated computation. Then, we follow the standard training process of the original 3D detectors, but only modify the data processing. After sampling the original images, we randomly insert the cached adversarial patch into them while keeping ground truth unchanged. This approach optimizes the detector to neglect the effects of adversarial attacks, thereby enhancing its robustness. In addition, we find that our adversarial training not only improves robustness but also enhances clean data performance, demonstrating the effectiveness of our method.
>
> **(2) About realism.**
>
> Thank you for pointing this out. We have conducted real-world experiments during the rebuttal phase. Please refer to the one-page PDF for more information. By printing the adversarial texture and adhering it to a vehicle model, the adversarial model successfully reduces the predicted confidence, demonstrating its practicality in real-world scenarios.
>
> **(3) Number of rendered images.**
>
> We use 10,000 rendered images for adversarial training. We use the hyperparameters in Section A of supplement material to control the location and rotation of objects. We do not use any heuristics except for the proposed primitive-aware sampling and semantic-guided regularization.
>
> **(4) Computational costs.**
>
> As we perform transferable attacks, our pipeline consists of two phases: training and inference. The training phase takes approximately two days using 8 NVIDIA A100 GPUs for 5 epochs in the nuScenes dataset. The inference phase of each frame, which involves rendering a single patch, takes around 0.2 seconds on an A100 GPU.

---

### Official Review · Reviewer_RUyN · 2023-07-09

**Soundness:** 2 fair
**Presentation:** 3 good
**Contribution:** 2 fair
**Rating:** 4
**Confidence:** 5

**Summary:**

Deep neural networks (DNNs) have shown susceptibility to adversarial examples, which raises significant safety concerns, particularly in safety-critical applications like DNN-based autonomous driving systems and 3D object detection. While there is a wealth of research on image-level attacks, most of them focus on the 2D pixel space, which may not always translate into physically realistic attacks in our 3D world. In this paper, the authors present Adv3D, the first exploration of modeling adversarial examples as Neural Radiance Fields (NeRFs).

The utilization of NeRFs allows for the generation of adversarial examples that possess photorealistic appearances and accurate 3D generation, thereby enabling more realistic and realizable adversarial attacks in the 3D domain. The authors train their adversarial NeRF by minimizing the confidence of surrounding objects predicted by 3D detectors on the training set. They evaluate Adv3D on an unseen validation set and demonstrate its ability to significantly degrade performance when rendering the NeRF in various sampled poses.

To ensure the practicality of the adversarial examples, the authors propose primitive-aware sampling and semantic-guided regularization techniques, which facilitate 3D patch attacks with camouflage adversarial textures. The experimental results showcase the generalizability of the trained adversarial NeRF across different poses, scenes, and 3D detectors. Additionally, the authors provide a defense mechanism against these attacks through adversarial training via data augmentation.

In summary, the authors introduce Adv3D as a novel approach that models adversarial examples using NeRFs, resulting in more realistic and realizable attacks in the 3D domain. They demonstrate the effectiveness of their method through extensive evaluations and propose a defense strategy to mitigate the impact of these attacks.

**Strengths:**

1. The writing and presentation of this paper are good and clear.
2. The idea of leveraging NeRF in generating adversarial examples is interesting
3. The authors conducted sufficient evaluation of the proposed method

**Weaknesses:**

1. The practicality of the proposed attack is questionable
2. There is a lack of real-world experiments
3. The study seems to lack technical insights as the adversarial attacks are well established. There is no surprise that using some fancy new idea can lead to adversarial examples, but the essence is the same.

**Questions:**

Please see the weakness section.

**Limitations:**

The authors has discussed the potential negative societal impact of this study, so it should be fine.

---

> ### Author Rebuttal · Authors · 2023-08-06
>
> We would like to thank the reviewer for the detailed feedback and for perceiving our methods as novel and effective. In the following, we reply to individual questions and comments raised by the reviewer:
>
> **(1,2) Real-world Experiments.**
>
> Thank you for pointing this out. We have conducted real-world experiments during the rebuttal phase. Please refer to the one-page PDF for more information. By printing the adversarial texture and adhering it to a vehicle model, the adversarial model successfully reduces the predicted confidence, demonstrating its practicality in real-world scenarios.
>
> **(3) Technical Insights.**
>
> We appreciate the feedback on our study. However, we respectfully disagree with the assessment that our work lacks technical insights. It is non-trivial to generate 3D adversarial examples using NeRF. We illustrate our contribution as follows:
>
> 1. Directly applying NeRF that modeling a whole scene as an adversarial example is impractical and difficult to realize in our real world. To provide a feasible attack, we propose primitive-aware sampling to enable 3D patch attacks that the adversarial NeRF only has a small modification to the original 3D environment. Furthermore, we introduce semantic-guided regularization that allows for a clear distinction between feasible and unfeasible areas. This enhances physical realizability by removing adversarial texture on infeasible areas, such as tires and wheels. In addition, our newly added real-world experiments also display satisfactory attack results (please see one-page PDF), proving the physical realizability and effectiveness of our method in practice.
>
> 2. To perform transferable attacks across poses and scenes, we formulate our learning objective as Expectation Over Transformation (EOT). The experimental results demonstrate that our method transfers well to different poses, unseen scenarios, and detectors in a non-contact manner. Additionally, we provide an adversarial defense method that not only improves robustness but also enhances clean data performance, demonstrating the effectiveness and benefits of our method.
>
> 3. We conduct extensive experiments to evaluate the robustness of different types of 3D detectors, including FoV and BEV, and provide a detailed analysis of each. This analysis may provide insightful implications for the development of more robust 3D detectors in the future. Specifically, in Section 5.3, we find that query-based detectors (DETR3D) are the most robust detectors, which provides valuable insights for building 3D detectors with enhanced robustness.
>
>
> We believe that the three aforementioned contributions collectively make a notable and insightful impact in the field.

---

> > ### Comment · Reviewer_RUyN · 2023-08-15
> > **Response**
> >
> > I would thank the authors for the rebuttal, especially for their efforts in creating real-world examples. However, the examples actually increase the concerns from the reviewer on the practicality of the attack. I would have to say that as a human, the confidence of the vehicle with the adversarial sticker to be a real one will drop in my mind. If I simply make the sticker with the same texture and color style of the road, the confidence may be even lower from the detector.
> >
> > Regarding the insights, I still think that the formulation of an adversarial attack using NeRF is not necessarily better than other adversarial attack formulations. I did not challenge the formulation itself, but the need to leverage NeRF for adversarial attacks. As mentioned before, I strongly believe a pattern of sticker that mimic the background objects (tree or building) will also lower the confidence of the prediction.
> >
> > So I will maintain my current score of 4.

---

> > > ### Author Response · Authors · 2023-08-16
> > > **Thank you for the feedback**
> > >
> > > Dear reviewer RUyN,
> > >
> > > We thank you for the comment. We appreciate the feedback. However, there may be a misunderstanding regarding the reviewer’s statement about how we attack the detector. Allow us first clarify our setting: our adversarial example aims to minimize the confidence of all surrounding objects (itself + others) in a non-contact manner. Simply making the sticker with the same texture and color style of the road will not lower the confidence of other untouched objects, as it would just blend the object into the background. Our adversarial examples, however, are effective in lowering the confidence of all surrounding objects, both in digital and real-world settings.
> > >
> > > Our adversarial attack pipeline is object-agnostic and can be adapted to any category of objects like trees or buildings. We chose vehicles as adversarial examples because they are the most common objects in driving scenarios. Evaluating and improving the robustness of 3D detectors based on vehicles can be the most effective approach.
> > >
> > > If there are any further concerns or questions, we would be happy to address them in further discussion.
> > >
> > > Thank you!

---

> > > > ### Comment · Reviewer_RUyN · 2023-08-16
> > > > **Discussion**
> > > >
> > > > I am aware of the adversarial objectives depicted in the paper, but this is not shown successfully in the real-world experiment. The lowered confidence is still hight for the surrounding cars.

---

> > > > > ### Author Response · Authors · 2023-08-16
> > > > > **Thank you for the discussion**
> > > > >
> > > > > Dear reviewer RUyN,
> > > > >
> > > > > Thank you for the comment. We agree that the attack effectiveness under real-world setting may not be fully exploited due to limited time and resources during the rebuttal phase. The inherent reasons that reduce our effectiveness in the real world can be the domain gap between the trained environment and our real-world environment, different camera parameters, and so on. We want to emphasize that our major goal and contribution is to leverage NeRF to better understand the robustness of 3D detectors in driving scenarios rather than crafting a real-world attacker. These understandings also contribute to our proposed training techniques (e.g., defense by data augmentation) to improve the clean performance and robustness of detectors, which is not necessary with a real-world attacker, as agreed by reviewer s7Sz. We believe that the real-world experiment is a bonus and is orthogonal to our core contribution.
> > > > >
> > > > > As the first exploration of modeling adversarial examples as NeRF, our goal is not to provide a powerful attack method, but aim to make an early attempt to investigate the effectiveness of NeRF in generating adversarial examples, which serve as a solid baseline in this field. On the other hand, the techniques of how to adapt adversarial NeRF in the real world might be worth to be another research paper.
> > > > >
> > > > > If there are any further concerns or questions, we would be happy to address them in further discussion.
> > > > >
> > > > > Thank you!

---

### Official Review · Reviewer_Xm3G · 2023-07-10

**Soundness:** 3 good
**Presentation:** 3 good
**Contribution:** 2 fair
**Rating:** 4
**Confidence:** 4

**Summary:**

The authors proposed new generative adversarial examples in the form of NeRFs, in the context of driving scenarios. The training objective is minimizing the 3D detection confidence from a variety of views. The parameters to optimize are the latent input to the NeRF, that encodes shape and texture info. Rendering is naturally differentiable due to the usage of NeRF. To improve the physical realizability, they propose three methods: primitive-aware sampling, NeRF disentanglement, and semantic-guided regularization. The authors conducted experiments on the widely used nuScenes dataset to evaluate the performance drop. The results show that their method is able to reduce the detection performance of various detectors, whether they are FOV-based detectors or birdview-based ones. They also evaluated the transferability of their method, and the adversarial training defense method.

**Strengths:**

Using NeRF as 3D adversarial example representation seems novel and interesting. The NeRF representation naturally is differentiable in terms of rendering, so it makes the adversarial attack problem easier. Also, with more uses of NeRF in 3D vision, it is important to explore the vulnerability in NeRF itself. Such adversarial attacks may highlight the potential security issues in NeRF.

The attacking framework (expectation over transformation), the NeRF rendering framework they use (Lift3D) are standard. The method is mostly built upon existing works; it seems not hard to implement their method.

The writing is clear.

**Weaknesses:**

My major concern is that whether the formulation of NeRF is necessarily, from the motivation perspective. In line 175, they fixed the shape and only optimized the texture latent code. The optimization is essentially finding the color, density of the volume. However, I believe most vehicle objectives are not translucent; the optimized 3D object is very hard to realize. This is evident as authors need to improve the physical realizability (line 180).

This leads to the Occam's razor principle: do we really need NeRFs to reach the effectiveness/realizability of the 3D attack? So we are missing a baseline here: optimizing the surface texture as a 3D mesh, using existing differentiable mesh renderers (such as Neural Mesh Renderer). The latter is easier to optimize (2D texture space), and more physically realizable (because it is a texture map rather than a volume). In line 166, the authors said "enables patch attacks in a 3D-aware manner by lifting the 2D patch to a 3D box", so we really need a baseline to showcase such lifting is necessary. Also it is not clear how rendering the NeRF into 3D scenes is done. In Fig. 3, the lighting of the NeRF object is not consistent with the environment, and we can see typical blurriness of NeRF.

Another weakness is that the setting is not sophisticated enough to be "Driving Scenarios". At first glance, it looks like attacking self-driving algorithms, but the point clouds are not used (correct me if I am wrong). The detection methods (FCOS3D, PGD-Det etc) are based on monocular/multi-view 2D images instead of multi-sensor. In Fig. 3, the inserted adversarial example does not seem to block the LiDAR rays. The experiment is not done through a full driving simulation software, but by rendering 3D objects into existing 3D data. Whether such mixed environment can represent the real-world driving scenario is not clear. It'll be better to claim general 3D detection scenario and do more experiments with other objects, instead of only claiming driving-specific scenarios.

In general, my decision largely depends on the first point: the NeRF representation may not be necessary under the current settings. Optimizing the texture image should just work; such volume formulation will make it harder to physically realize and does not bring much benefit other than differentiable rendering.

**Questions:**

1. How does the method to a naive baseline, such as direct optimization of the texture of the car? Is it possible to do any simple ablation experiments?
2. Is it possible to reconstruct the NeRF, 3D-print it, then test in the real world?
3. How is the NeRF rendered into nuScenes? There are shadows under the NeRF renderings, but somehow the lighting is not consistent. Is it because the NeRF formulation omits shading (line 119)? Can you clarify?

**Limitations:**

The authors addressed the limitations about dataset annotations and potential harmful consequences.

---

> ### Author Rebuttal · Authors · 2023-08-06
>
> We would like to thank the reviewer for the detailed feedback and for appreciating the novelty of our idea and clear writing. In the following, we reply to individual questions and comments raised by the reviewer:
>
>
> **(1) Is it necessary to use NeRF as the representation of adversarial examples?**
>
> We acknowledge that other 3D representation like mesh has its own advantages as an adversarial example, such as having a clear surface definition. However, NeRF also has its distinguished advantages that deserve to explore for crafting adversarial examples:
>
> 1. **Latent representation.** The high-dimensional latent space of NeRF provides greater flexibility for downstream applications. In Section C of the supplementary material, we demonstrate that given the latent embedding of an adversarial texture, we can easily transfer the trained texture to other previously unseen vehicles while preserving fidelity. In contrast, transferring adversarial texture to unseen vehicles using mesh can be challenging.
>
>
>
>
> 2. **NeRF vs Mesh.** As suggested, we conduct a comparison of NeRF attack and mesh attack. We provide a simple experiment using a randomly picked ShapeNet car model as a mesh baseline. We use PyTorch3D’s differentiable renderer and optimized the vertex color as an adversarial example to attack 3D detectors. The attack performance of mAP and NDS is slightly lower than the NeRF counterpart. This may be attributed to the latent space of NeRF network has a higher dimensional representation than vertex color and providing much more solutions for attacking, which results in a better attack effect. This experiment also proves the advantage of using latent representation for adversarial examples.
> | Method      | NDS | mAP |
> | ----------- | ----------- | ----------- |
> | Clean      | 0.3822 | 0.3076 |
> | Mesh attack   | 0.3018 | 0.2183 |
> | NeRF attack   | 0.2648 | 0.1895 |
>
>
> 3. **Physical realizability of NeRF.** Our method defines the NeRF volume in SDF (Signed Distance Field) space, converging the volume to a surface area on the zero-level set, thus the volume has a meaningful definition. Our real-world experiments (please see one-page PDF) also demonstrate that the adversarial texture can be created in the real world and display satisfactory attack results.
>
>
>
>
> 4. **Realism.** In some cases, NeRF is more realistic than meshes (e.g., Lift3D vs ShapeNet model). This is because NeRF is created from real-world captured data, while meshes often require artists to manually adjust vertices, textures, and lighting, which sometimes suffer from domain gaps. Therefore, we hypothesize that **1.** Improving the robustness of detectors based on the mesh may not guarantee improvement in the real world. **2.** The adversarial textures trained using NeRF (especially under semantic-guided regularization) may generalize better to the real world.
>
>
> 5. **Broader impact.** The recent development of NeRF has led to remarkable progress in NeRF-based driving scene simulation [a,b,c]. Our adversarial framework is general and can be extended to integrate with the advances in NeRF-based simulators to benefit a wide spectrum of practical systems. For instance, our framework can be combined with UniSim [a] to perform adversarial closed-loop evaluations of self-driving cars in NeRF environments, or with ClimateNeRF [b] to identify adverse weather conditions that may corrupt the autonomous driving system. We believe that our work provides valuable insights and opens up new possibilities for creating authentic adversarial evaluations that improve the robustness of self-driving cars.
>
>
>
> **(2) Point cloud generation.**
>
> It does not take much effort to simultaneously generate images and LiDAR point clouds using NeRF as described in [a]. In our work, we chose to only evaluate the most common and challenging modality, which is images for brevity. We leave the multi-modality adversarial evaluation for future exploration.
>
> **(3) Claim issue: "Driving Scenarios".**
>
> Thanks for pointing this out. It would be more clear and accurate that use "3D object detection" than "Driving Scenarios". We will revise our claim accordingly.
>
> **(4) Question 1: Comparision.**
>
> In Tab. 3 of the main paper, we provide a comparison of full texture optimization and semantic-guided regularization. The attacked mAP is $0.132$ versus $0.148$. As expected, reducing the area of adversarial parts slightly decreases the attack performance.
>
> **(5) Question 2: Real-world experiments.**
>
> Yes. We have added real-world experiments during the rebuttal phase. Please refer to the one-page PDF for more information. By printing the adversarial texture on A4 paper and adhering it to a vehicle model, the adversarial model successfully reduces the predicted confidence, demonstrating its practicality in real-world scenarios.
>
> **(6) Question 3: Rendering issue.**
>
> Our NeRF follows the standard paradigm that omits shading. We find it sufficient to cast shadows from a pre-computed shadow map. Although lighting estimation is vital for physically-based rendering, it may be out of the scope of this paper. Future work can leverage [d] to perform accurate shadow casting.
>
> **References**
>
> [a] UniSim: A Neural Closed-Loop Sensor Simulator, CVPR 2023
>
> [b] ClimateNeRF: Physically-based Neural Rendering for Extreme Climate Synthesis, ICCV 2023
>
> [c] Lift3D: Synthesize 3D Training Data by Lifting 2D GAN to 3D Generative Radiance Field, CVPR 2023
>
> [d] Neural Fields meet Explicit Geometric Representations for Inverse Rendering of Urban Scenes, CVPR 2023

---

### Official Review · Reviewer_R8Y1 · 2023-07-13

**Soundness:** 4 excellent
**Presentation:** 3 good
**Contribution:** 3 good
**Rating:** 5
**Confidence:** 4

**Summary:**

This work proposes to generate 3D adversarial examples for attacking 3D object detectors in driving scenarios using NeRF. In particular, it integrates a series of techniques, including primitive-aware sampling and semantic-guided regularization, to ensure the physical realism and realizability of the generated adversarial examples. Extensive experiments have validated the effectiveness of the proposed method in reducing detection performance and serving as data augmentation.

**Strengths:**

1. As an early attempt of generating 3D adversarial examples using NeRF, this work could offer a new perspective for the community in understanding and tackling real-world 3D adversarial attacks.

2. The extensive experiments validate the superiority of NeRF as a 3D adversarial attack generator. In particular, it is interesting to see that the generated adversarial examples can serve as a data augmentation to improve clean performance, which aligns with the previous observations in classification.

**Weaknesses:**

1. My major concern is the assumed attacking setting of this work, i.e., how to leverage the proposed method in real-world driving scenarios. If only a static adversarial example is attached to the scene, generating other static objects on the road may be more practical than generating a vehicle; Otherwise, the authors are expected to show a video under an egocentric view to demonstrate the attack effectiveness, i.e., whether the dynamically moving adversarial vehicles can consistently mislead the 3D detectors from different view directions.

2. The claim "the first exploration of modeling adversarial examples as Neural Radiance Fields (NeRFs)" in the abstract may not be accurate. ViewFool [1] also models adversarial examples using NeRF although only the view direction is adversarially optimized. It will be more accurate if the authors highlight this work as the first 3D adversarial example generator using NeRF.

3. Missing references regarding the early attempts of marrying NeRF and adversarial attacks (which are mostly orthogonal with this work):

[1] "ViewFool: Evaluating the Robustness of Visual Recognition to Adversarial Viewpoints", Y. Dong et al., NeurIPS'22.

[2] "NeRFool: Uncovering the Vulnerability of Generalizable Neural Radiance Fields against Adversarial Perturbations", Y. Fu et al., ICML'23.

[3] "Aug-NeRF: Training Stronger Neural Radiance Fields With Triple-Level Physically-Grounded Augmentations", T. Chen et al., CVPR'22.

4. Minor issue: There exists some inconsistency in terms of tense and punctuation, which could be improved in the final version.

**Questions:**

1. In my understanding, the primitive-aware sampling essentially performs a coordinate transformation between the world coordinate of the driving scenario and the canonical space defined by Lift3D. Will this transformation stretch the shape of the generated adversarial vehicles?

2. How could you generate adversarial vehicles in different styles if the mapping between the optimized texture code and the vehicle texture is deterministic?

**Limitations:**

Although the developed adversarial attacks may cause security concerns, this work intended to gain a deeper understanding of 3D adversarial examples and improve the achievable robustness on them, thus not suffering from negative societal impact.

---

> ### Author Rebuttal · Authors · 2023-08-06
>
> We would like to thank the reviewer for the positive and detailed feedback. Below, we reply to individual comments and questions raised by the reviewer:
>
> **(1) Real-world experiments.**
>
> It is practicable to produce an adversarial NeRF in the real world by printing an adversarial texture. Our added experiments in the real world (see one-page PDF) show that by rendering the orthogonal view of the texture and simply pasting it to vehicle models, we achieve reasonable results to attack 3D detectors. A more advanced method can leverage recent work [a] to extract the underlying texture of NeRF.
>
> **Sequential results.** Providing sequential results such as video is valuable feedback. To evaluate the effects of the movement of objects, we present experiments in Fig. 4 of the main paper, which demonstrates the effectiveness of our attack across different locations and rotations.
>
> **(2, 3) Claim issue and references.**
>
> Thank you for pointing this out. The NeurIPS 2022 paper "ViewFool" leverages the differentiability of NeRF to find adversarial view directions. This work is related to ours, and we will add a discussion about it and revise our claim accordingly. "Aug-NeRF" investigates the robustness of NeRF reconstruction itself but does not model adversarial examples as NeRF. "NeRFool" does a similar thing to "Aug-NeRF" and is available online **after** the NeurIPS 2023 deadline. Compared with these works, our method uses NeRF as an adversarial example generator. We thoroughly evaluate the robustness of 3D detectors by leveraging NeRF's photorealistic synthesis and differentiability, and provide insightful analysis to develop more robust 3D detectors. We will cite them and add a comparison in our related work.
>
> **(4) Improvement of writing.**
>
> Thanks for pointing out the tense and punctuation issues. We will revise our paper further.
>
> **(5) Q1: Whether stretching.**
>
> Yes. In our implementation, we randomly stretch the length, width, and height of the adversarial vehicles to enhance transferability. Detailed hyperparameters can be found in Section A of our supplementary material.
>
> **(6) Q2: Transfer to unseen vehicle.**
>
> The optimized texture code and the vehicle texture are not tightly coupled. We can easily transfer the adversarial texture to unseen shape of vehicle (please see the visualization in Section F of the supplement material).
>
> **References**
>
> [a] Delicate Textured Mesh Recovery from NeRF via Adaptive Surface Refinement, ICCV 2023

---

> ### Comment · Reviewer_R8Y1 · 2023-08-15
> **Reviewer Response**
>
> Thank the authors for the great efforts made by the rebuttal and the real-world experiments look interesting. However, the attack effectiveness under a real-world setting may be limited (which can be caused by many non-technical reasons in the uncertain real world) and it is currently hard to say whether the perturbation is an adversarial attack or just a strong noise. As such, I will keep my original score for now and I am willing to discuss it with other reviewers to further adjust my scores.

---

> > ### Author Response · Authors · 2023-08-16
> > **Thank you for the feedback!**
> >
> > Dear reviewer R8Y1,
> >
> > Thank you for providing valuable comments. We agree that the attack effectiveness under real-world setting may not be fully exploited due to limited time and resources during the rebuttal phase. The inherent reasons that reduce our effectiveness in the real world can be the domain gap between the trained environment and our real-world environment, different camera parameters, lighting issue, and so on.  We want to emphasize that our major goal and contribution is leveraging adversarial NeRF to better understand the robustness of 3D detectors in driving scenarios rather than crafting a real-world attacker. These understandings also contribute to our proposed training techniques (e.g., defense by data augmentation) to improve the clean performance and robustness of detectors, which is not necessary with a real-world attacker, as agreed by reviewer s7Sz. We believe that the real-world experiment is a bonus and is orthogonal to our core contribution.
> >
> > **Adversarial NeRF *vs* Strong Noise.** To further address the reviewer's concern, we conduct additional experiments comparing adversarial NeRF with strong noise in real-world settings. In these experiments, we replace the adversarial texture area with various types of noise, including pure black, the mean color of the background image, and random noise. In the table below, we observe that adversarial NeRF achieves the lowest predicted confidence and outperforms the other three types of texture in terms of attack effectiveness for hiding surrounding objects.
> >
> > | Texture | Clean | Pure Black | Mean Color | Random Noise | Adversarial NeRF |
> > | :-: | :-: | :-: | :-: | :-: | :-: |
> > | Confidence | 0.672 | 0.632 | 0.634 | 0.644 | **0.625** |
> >
> > We thank the reviewer for providing the feedback! If the reviewer has any further questions or suggestions, we are more than happy to take them.

---

### Author Rebuttal · Authors · 2023-08-07

We would like to thank all the reviewers for their insightful reviews. Before addressing the specific questions in the individual replies, we would like to first reiterate our motivation and contribution, and then provide a detailed description of the experiments that we have added during the rebuttal phase.


**(1) Motivation and Contribution**

Given the safety-critical demand for self-driving cars, it is critical to gain a deeper understanding of the robustness of 3D detectors in driving scenarios using 3D adversarial examples like NeRF. However, it is non-trivial to apply NeRF in adversarial attacks. We illustrate our contribution as follows:

1. Directly applying NeRF that modeling a whole scene as an adversarial example is impractical and difficult to realize in our real world. To provide a feasible attack, we propose primitive-aware sampling to enable 3D patch attacks that the adversarial NeRF only has a small modification to the original 3D environment. Furthermore, we introduce semantic-guided regularization that allows for a clear distinction between feasible and unfeasible areas. This enhances physical realizability by removing adversarial texture on infeasible areas, such as tires and wheels. In addition, our newly added real-world experiments also display satisfactory attack results (please see one-page PDF), proving the physical realizability and effectiveness of our method in practice.

2. To perform transferable attacks across poses and scenes, we formulate our learning objective as Expectation Over Transformation (EOT). The experimental results demonstrate that our method transfers well to different poses, unseen scenarios, and detectors in a non-contact manner. Additionally, we provide an adversarial defense method that not only improves robustness but also enhances clean data performance, demonstrating the effectiveness and benefits of our method.

3. We conduct extensive experiments to evaluate the robustness of different types of 3D detectors, including FoV and BEV, and provide a detailed analysis of each. This analysis may provide insightful implications for the development of more robust 3D detectors in the future. Specifically, in Section 5.3, we find that query-based detectors (DETR3D) are the most robust detectors, which provides valuable insights for building 3D detectors with enhanced robustness.


**(2) Broader Impact**

The recent development of NeRF has led to remarkable progress in NeRF-based driving scene simulation [a,b,c]. Our adversarial framework is general and can be extended to integrate with the advances in NeRF-based simulators to benefit a wide spectrum of practical systems. For instance, our framework can be combined with UniSim [a] to perform adversarial closed-loop evaluations of self-driving cars in NeRF environments, or with ClimateNeRF [b] to identify adverse weather conditions that may corrupt the autonomous driving system. We believe that our work provides valuable insights and opens up new possibilities for creating authentic adversarial evaluations that improve the robustness of self-driving cars.

**(3) Additional Experiments**


1. **Real-World Experiments.** To validate the practicality of our adversarial example, we conduct experiments using scaled models (1:24) of real-world vehicles (see one-page PDF). We approximate adversarial textures using the rendering of the orthogonal views of examples (Future work can leverage [d] to extract the exact texture of NeRF). Next, we print the adversarial texture on A4 paper and tailor it to fit our vehicle model. Our experiments show that the adversarial texture is successful in reducing the confidence of both itself and surrounding objects, proving the practicality of our adversarial example.
2. **Mesh Comparison.** As suggested by reviewer Xm3G and s7Sz, we provide a simple experiment using a randomly picked ShapeNet car model as a mesh baseline. We use PyTorch3D’s differentiable renderer and optimized the vertex color as an adversarial example to attack 3D detectors (BEVDet). Align with the setting of the NeRF counterpart, we randomly rendered the mesh model and pasted the patch onto the original images. The attack performance of mAP and NDS is slightly lower than the NeRF counterpart. This may be attributed to the latent space of NeRF network has a higher dimensional representation than vertex color and providing much more solutions for attacking, which results in a better attack effect.
| Method      | NDS | mAP |
| ----------- | ----------- | ----------- |
| Clean      | 0.3822 | 0.3076 |
| Mesh attack   | 0.3018 | 0.2183 |
| NeRF attack   | 0.2648 | 0.1895 |


**References**

[a] UniSim: A Neural Closed-Loop Sensor Simulator, CVPR 2023

[b] ClimateNeRF: Physically-based Neural Rendering for Extreme Climate Synthesis, ICCV 2023

[c] Lift3D: Synthesize 3D Training Data by Lifting 2D GAN to 3D Generative Radiance Field, CVPR 2023

[d] Delicate Textured Mesh Recovery from NeRF via Adaptive Surface Refinement, ICCV 2023

---

### Decision · Program_Chairs · 2023-09-21

**Decision:**

Reject

**Comment:**

This is a borderline paper. AC read all reviewers' comments and the rebuttal. The main concern left is the practicality as pointed out by Reviewer RUyN and Xm3G. Moreover, from the experiments request by Reviewer Xm3G, the improvement is not that high compared to mesh-based baseline.  After the careful discussion with reviewers, AC feels that the quality of this paper can be further improved based on the reivewer's comments.